# A Combination of Alectinib and DNA-Demethylating Agents Synergistically Inhibits Anaplastic-Lymphoma-Kinase-Positive Anaplastic Large-Cell Lymphoma Cell Proliferation

**DOI:** 10.3390/cancers15205089

**Published:** 2023-10-21

**Authors:** Kazunori Kawasoe, Tatsuro Watanabe, Nao Yoshida-Sakai, Yuta Yamamoto, Yuki Kurahashi, Keisuke Kidoguchi, Hiroshi Ureshino, Kazuharu Kamachi, Yuki Fukuda-Kurahashi, Shinya Kimura

**Affiliations:** 1Department of Drug Discovery and Biomedical Sciences, Faculty of Medicine, Saga University, Saga 849-8501, Japan; 2Division of Hematology, Respiratory Medicine and Oncology, Department of Internal Medicine, Faculty of Medicine, Saga University, Saga 849-8501, Japan; 3OHARA Pharmaceutical Co., Ltd., Koka 520-3403, Japan

**Keywords:** ALK, ALCL, alectinib, DNA-demethylating agents, synergistic efficacy

## Abstract

**Simple Summary:**

Tyrosine kinase inhibitors (TKIs) targeting anaplastic lymphoma kinase (ALK), such as crizotinib and alectinib, provide favorable clinical responses in human malignancies driven by ALK fusion proteins, including ALK-positive anaplastic large cell lymphoma (ALK^+^ ALCL) and non-small cell lung carcinoma (ALK^+^ NSCLC). ALK TKIs were approved for clinical use in ALK^+^ NSCLC patients prior to being approved for use in individuals with ALK^+^ ALCL. Although most ALK^+^ NSCLC patients initially respond to crizotinib and alectinib therapy, they relapse after several years on this therapy. Here, we investigated whether a combination of alectinib and DNA-demethylating agents could have synergistic efficacy for the treatment of ALK^+^ ALCL patients. We found that the combination of alectinib and OR-2100, an orally bioavailable decitabine prodrug, synergistically suppressed ALCL cell proliferation, which was accompanied by gene expression reprograming. Therefore, alectinib and OR-2100 combination therapy has the potential to improve treatment outcomes in patients with ALK^+^ ALCL.

**Abstract:**

The recent evolution of molecular targeted therapy has improved clinical outcomes in several human malignancies. The translocation of anaplastic lymphoma kinase (ALK) was originally identified in anaplastic large-cell lymphoma (ALCL) and subsequently in non-small cell lung carcinoma (NSCLC). Since *ALK* fusion gene products act as a driver of carcinogenesis in both ALCL and NSCLC, several ALK tyrosine kinase inhibitors (TKIs) have been developed. Crizotinib and alectinib are first- and second-generation ALK TKIs, respectively, approved for the treatment of ALK-positive ALCL (ALK^+^ ALCL) and ALK^+^ NSCLC. Although most ALK^+^ NSCLC patients respond to crizotinib and alectinib, they generally relapse after several years of treatment. We previously found that DNA-demethylating agents enhanced the efficacy of ABL TKIs in chronic myeloid leukemia cells. Moreover, aberrant DNA methylation has also been observed in ALCL cells. Thus, to improve the clinical outcomes of ALK^+^ ALCL therapy, we investigated the synergistic efficacy of the combination of alectinib and the DNA-demethylating agent azacytidine, decitabine, or OR-2100 (an orally bioavailable decitabine derivative). As expected, the combination of alectinib and DNA-demethylating agents synergistically suppressed ALK^+^ ALCL cell proliferation, concomitant with DNA hypomethylation and a reduction in STAT3 (a downstream target of ALK fusion proteins) phosphorylation. The combination of alectinib and OR-2100 markedly altered gene expression in ALCL cells, including that of genes implicated in apoptotic signaling, which possibly contributed to the synergistic anti-ALCL effects of this drug combination. Therefore, alectinib and OR-2100 combination therapy has the potential to improve the outcomes of patients with ALK^+^ ALCL.

## 1. Introduction

Anaplastic large-cell lymphoma (ALCL) is a rare type of non-Hodgkin’s lymphoma (NHL) comprising four subtypes: anaplastic lymphoma kinase (ALK)-positive ALCL (ALK^+^ ALCL), ALK-negative ALCL (ALK^−^ ALCL), primary cutaneous ALCL (pcALCL), and breast-implant-associated ALCL (BIA-ALCL) [1]. ALCL accounts for ~10–15% and ~1–2% of pediatric and adult NHL cases, respectively [2]. ALK^+^ ALCL is the most common form of ALCL, constituting ~90% of pediatric and ~50% of young adult cases [2]. ALK is a receptor tyrosine kinase that was originally discovered as a component of the ALK–nucleophosmin 1 (NPM1) fusion protein generated via the t(2;5)(p23;q35) chromosomal translocation [3]. NPM1-ALK is an intracellular chimeric protein with ligand-independent constitutive tyrosine kinase activity [4]. It activates several downstream signaling pathways, including JAK3/STAT3, Ras/MAPK/ERK, and PI3K/AKT/mTOR, imparting a growth advantage on tumor cells [4]. Although NPM1-ALK is a major ALK fusion protein found in patients with ALCL (accounting for 80% of all ALK^+^ ALCL cases), several other ALK fusion partners have been discovered, including tropomyosin 3 (TPM3), 5-aminoimidazole-4-carboxamide ribonucleotide (ATIC), and TRK-fused gene (TFG) [4].

The CHOP combination chemotherapy regimen, which consists of cyclophosphamide, doxorubicin, vincristine, and prednisone, is widely used as the first line treatment for ALK^+^ ALCL. More recently, the CD30-targeting agent brentuximab vedotin was incorporated into the CHOP formulation. The overall survival and progression-free survival (PFS) rates after 5 years for ALK^+^ ALCL undergoing CHOP treatment are over 70% and 60%, respectively, with 30–40% patients experiencing relapse [1]. Although allogenic hematopoietic stem cell transplantation is one therapeutic approach for relapsed and refractory ALK^+^ ALCL, molecular targeted therapies have been established or are under development. Similar to the discovery of *NPM1-ALK* in ALCL [3], the echinoderm microtubule-associated protein-like 4 (*EML4)-ALK* fusion gene was discovered as a cancer driver gene, which is detected in ~7% of patients with non-small cell lung carcinoma (NSCLC) [5]. Subsequently, an ALK-targeting tyrosine kinase inhibitor (TKI) has been developed for the treatment of ALK^+^ NSCLC.

Crizotinib is a first generation ALK TKI that targets ALK, c-MET [6], and ROS1 [7], which was approved by the Food and Drug Administration (FDA) for ALK^+^ NSCLC in 2011. Subsequently, the efficacy of crizotinib against ALK^+^ ALCL was reported in several clinical trials [8,9], leading to its approval as a therapy for young adult or pediatric patients with relapsed or refractory ALCL in 2021. In comparison with crizotinib, the second-generation ALK TKI alectinib shows improved specificity for ALK and superior efficacy against EML-ALK harboring the crizotinib-resistant mutation L1196M, which is detected in patients with crizotinib-relapsed/refractory NSCLC [10]. Furthermore, alectinib showed more favorable PFS and adverse event profiles than crizotinib in a phase 3 head-to-head comparison trial in Japanese patients with ALK-inhibitor-naïve ALK^+^ NSCLC [11]. Alectinib also showed favorable clinical activity in patients with ALK^+^ ALCL who had failed to respond to standard chemotherapy [12]. This led to alectinib being approved by the Ministry of Health, Labour and Welfare of Japan for the treatment of recurrent or refractory ALK^+^ ALCL in 2020.

Most ALK^+^ NSCLC patients respond to crizotinib and alectinib, but typically relapse within several years due to acquired resistance [13]. Although few published reports of ALCL cases exist, owing to the rarity of this condition, it is important to improve the efficacy of ALK TKIs in the treatment of patients with ALK^+^ ALCL.

An aberrant DNA methylation profile is observed in human malignancies and is a known target in cancer therapy [14]. DNA methyltransferase 1 (DNMT1) catalyzes the transfer of the methyl group from S-adenosyl methionine to a cytosine residue to form 5-methylcytosine, thus maintaining the DNA methylation status during DNA replication [15]. STAT3, which is a downstream target of the NPM1-ALK fusion protein, induces the transcription of DNMT1 [16], leading to its overexpression in ALK^+^ ALCL cells [17]. Furthermore, knocking out *DNMT1* clearly abrogates NPM1-ALK-induced tumorigenesis in transgenic mice carrying the human *NPM1-ALK* fusion gene [18]. Thus, DNA methylation might be closely associated with the leukemogenesis of ALK^+^ ALCL. In fact, tumor-cell-specific regional DNA hypermethylation is observed in both ALK^+^ ALCL and ALK^–^ ALCL [19]. The anti-apoptotic gene *MCL1* is constitutively highly expressed in ALK^+^ ALCL [20,21] as a result of miR-29a silencing via promoter hypermethylation [22]. Moreover, other tumor suppressors, such as SHP1 [23] and miR-150 [24], are also silenced in ALK^+^ ALCL cells via DNA methylation. 

Since an aberrant DNA methylation status is related to leukemogenesis in ALK^+^ ALCL, DNA hypomethylating therapy seems to be a convincing approach to treating ALK^+^ ALCL. A few studies have reported the efficacy of DNA-demethylating agents, such as azacytidine (AZA) and decitabine (DAC), in ALK^+^ ALCL cell lines [17,23,24]. Recently, we found that the combination of ABL1 TKIs with OR-2100 (OR21), which we developed as the first orally available, single-compound DAC prodrug [25,26], induced synergistic anti-leukemic effects in a chronic myeloid leukemia mouse model [27]. OR21 was less hematotoxic than DAC, but exhibited almost identical tumor growth inhibition to that of DAC in several mouse models [26,28]. A phase I clinical trial of OR21 (Japan Registry Clinical Trials number, jRCT2071220035) in patients with high-risk myelodysplastic syndromes is underway. 

Therefore, in the present study, we investigated the efficacy of the combination of ALK TKIs with DNA-demethylating agents including AZA, DAC, and OR21, for the treatment of ALCL.

## 2. Material and Methods

### 2.1. Reagents

AZA and DAC were purchased from Sigma-Aldrich (St. Louis, MO, USA). OR-2100 (OR21) was provided by OHARA Pharmaceutical Co. (Koka, Japan). Crizotinib and alectinib were purchased from Selleck Chemicals (Houston, TX, USA). All reagents were dissolved in DMSO and stored at −20 °C. 

### 2.2. Cell Culture

Four different human cell lines derived from patients with ALK+ ALCL were used. The SU-DHL-1, L-82, and SR-786 cells lines were sourced from DSMZ. The Karpas299 cell line was kindly provided by Kazuaki Yokoyama (University of Tokyo, Tokyo, Japan). Human primary hepatocytes were purchased from JCRB Cell Bank (Ibaraki, Japan). All cell lines were cultured in RPMI-1640 (Sigma-Aldrich) containing 10% fetal bovine serum.

### 2.3. Cell Proliferation Assay

The proliferation of ALCL cell lines was determined after 4 days of culturing in the presence of compounds using the Cell Counting Kit-8 (Dojindo Molecular Technology, Kumamoto, Japan).

### 2.4. LINE1 Methylation Assay by Bisulfite Pyrosequencing

The degree of methylation at CpG sites in the promoter-proximal region of *LINE-1* was measured using a pyrosequencing-based assay, as described previously [26]. Bisulfite-treated DNA was prepared using the EZ DNA Methylation Kit (Zymo Research, Irvine, CA, USA) and amplified via PCR. The PCR products were sequenced using the Pyromark Q24 (Qiagen, Hilden, Germany) and PyroMark Gold-Q24 (Qiagen) reagent kits.

### 2.5. Determining the Synergistic Anti-Tumor Effect of the Combination Treatment

ALCL cell lines were treated with five concentrations of alectinib, crizotinib, or DNA-demethylating agents (AZA, DAC, and OR21) for 4 days (5 × 5, for a total 25 conditions in each combination). The synergy scores were determined using the SynergyFinder (version 3.0) web application [29] with the ZIP calculation method [30].

### 2.6. Apoptosis Assays

Karpas299 cells were incubated for 4 days with various concentrations of alectinib with or without DNA-demethylating agents, and then stained with APC-conjugated Annexin V (BioLegend, San Diego, CA, USA) and propidium iodide (Sigma-Aldrich). Apoptotic cells were acquired using a FACSVerse cytometer (BD Biosciences, Franklin Lakes, NJ, USA), and the data were analyzed using FlowJo software version 10 (Tree Star, Ashland, OR, USA).

### 2.7. Cell Cycle Analysis

Karpas299 cells were incubated for 4 days with various concentrations of alectinib with or without DNA-demethylating agents, and then fixed with cold 70% ethanol. Fixed cells were incubated with 0.25 mg/mL RNase A (Qiagen) and stained with 50 µg/mL propidium iodide (Sigma-Aldrich). All samples were evaluated via flow cytometry and analyzed using FlowJo software version 10.

### 2.8. Western Blotting

Primary antibodies against phosphor-STAT3 (Tyr705), STAT3, and tubulin were purchased from Cell Signaling Technology (Danvers, MA, USA). The secondary antibody, HRP-conjugated donkey anti-rabbit IgG, was purchased from Cytiva (Marlborough, MA, USA). Cell lysates were prepared using a radioimmunoprecipitation assay buffer (Santa Cruz Biotechnology, Dallas, TX, USA) and subsequently resolved on Nu-polyacrylamide gels (Invitrogen, Waltham, MA, USA). Proteins were detected using antibodies and ECL reagents (Cytiva).

### 2.9. RNA Sequencing (Seq) and Analysis

Total RNA was isolated from Karpas299cells, after 4 days of treatment with alectinib with or without DNA-demethylating agents, using the Direct-zol RNA MiniPrep kit (Zymo Research). RNA-seq library preparation, sequencing on the NovaSeq 6000 system (Illumina, San Diego, CA, USA), and sequencing data analysis (performed via Rhelixa Inc., Tokyo, Japan) have been described previously [31]. Gene ontology (GO) analysis was performed using The Database for Annotation, Visualization and Integrated Discovery (DAVID) [32,33].

### 2.10. Xenograft Mouse Model

Animal studies were performed according to a protocol approved by Saga University (A2023-010-0) in accordance with the German Animal Welfare Act. Karpas299 cells were subcutaneously inoculated into 6-week-old female NOD/Shi-scid, IL-2Rγ KO Jic (NOG) mice (In-Vivo Science Inc., Kawasaki, Japan). A visible tumor appeared ten days after inoculation and the mice were randomized into four groups to obtain equal tumor sizes in each group (Vehicle, OR21, Alectinib, and OR21 + Alectinib). Tumor volumes were defined as (short axis)^2^ × (long axis)/2 and measured twice per week. The mice were treated with vehicle (10% hydroxypropyl-β-cyclodextrin (HP-β-CD) containing 1% dimethyl sulfoxide, intraperitoneally, twice per week and 0.02N HCL, 10% DMSO, 10% CremophorEL, 15% PEG400, and 15% HP-β-CD, oral gavage, daily, *n* = 11), OR21 (2 mg/kg, intraperitoneally, twice a week, *n* = 11), alectinib (2 mg/kg, oral gavage, daily, *n* = 11), or OR21 + alectinib (2 mg/kg OR21 and 2 mg/kg alectinib, *n* = 11). All mice were euthanized 13 days after the first treatment. Cell counts in peripheral blood were determined using an MEK-6500 Celltaca (Nihon Kohden, Tokyo, Japan).

### 2.11. Statistics

Data were expressed as the mean ± standard deviation (SD). Differences between groups were assessed using the Dunnett test and Tukey test. 

## 3. Results

### 3.1. Efficacy of DNA-Demethylating-Agent Monotherapy at Inhibiting ALCL Cell Growth In Vitro 

We first compared the efficacy of OR21 with that of the conventional DNA-demethylating agents, AZA and DAC, in ALCL cell lines in vitro. Treatment with AZA for 4 days inhibited cell growth in all four ALCL cell lines in a dose-dependent manner and to a similar degree; the 50% inhibitory concentration (IC_50_) values for SU-DHL-1, L-82, SR-786, and Karpas299 were 0.75, 0.85, 0.75, and 1.58 μM, respectively (Figure 1a). Although DAC also suppressed cell growth in a dose-dependent manner, it was less effective at inhibiting the growth of the SR-786 and Karpas299 cell lines than that of the SU-DHL-1 and L-82 cell lines; the IC_50_ values for SU-DHL-1, L-82, SR-786, and Karpas299 were 0.14, 0.08, 5.57, and 10 μM, respectively (Figure 1a). The treatment of primary human hepatocytes with AZA, DAC, and OR21 suppressed cell growth to some extent, possibly because the cells proliferated slowly (Figure 1a). The growth-inhibitory effect of OR21 in each cell line was almost the same as that of DAC (Figure 1a). We next measured the DNA methylation status of the *LINE-1* promoter region, a surrogate maker for global DNA methylation status [34]. The level of *LINE-1* promoter methylation was ~80% in all four cell lines (Figure 1b). Although the treatment of cells with AZA, DAC, or OR21 for four days reduced *LINE-1* promoter methylation, this decrease in methylation was not related to the extent of cell growth inhibition (Figure 1b).

### 3.2. Efficacy of ALK Tyrosine Kinase Inhibitor Monotherapy at Inhibiting ALCL Cell Growth In Vitro

We next examined the efficacy of the first-generation ALK TKI crizotinib and the second-generation ALK TKI alectinib in ALCL cell lines in vitro. The cell lines were cultured in the presence of crizotinib or alectinib for 4 days. Although the IC_50_ values for crizotinib were ~70 nM in SU-DHL-1, L-82, and Karapas299 cells, the SR-786 cells were less sensitive to its effects (IC_50_ = 225 nM) (Figure 2a). Alectinib inhibited cell growth more effectively than crizotinib. The IC_50_ values were in the 11–25 nM range in all cell lines, except for SR-786 (IC_50_ = 97 nM), which was less sensitive to its effects (Figure 2a). Both crizotinib and alectinib had little efficacy against normal hepatocytes (Figure 2a). The phosphorylation of STAT3, a well-known downstream target of the NPM1-ALK fusion protein, was reduced via treatment with crizotinib or alectinib in a dose-dependent manner (Figure 2b). Furthermore, the decrease in STAT3 phosphorylated was related to the amount of cell growth inhibition (Figure 2b).

### 3.3. DNA-Demethylating Agents Synergize with Alectinib to Inhibit the Growth of ALCL Cells

Since alectinb was more effective than crizotinib at inhibiting the growth of ALK^+^ ALCL cell lines in our in vitro experiments (Figure 2), we decided to focus on it for the remainder of the study and test it in combination with DNA-demethylating agents. Hence, the Karpas299 and SR-786 cell lines (Figure 3a) and the other two ALK^+^ ALCL cell lines (Appendix A) were treated with different concentrations of the DNA-demethylating agents (AZA, DAC, or OR21) or alectinib, either alone or with alectinib in combination with each of the DNA-demethylating agents. We subsequently calculated the magnitude of synergistic growth-inhibitory effects, which was expressed as a ZIP score using the SynergyFinder web application. The combination of a DNA-demethylating agent and alectinib induced synergistic effects (ZIP score > 10) at some of the concentration ranges in all four of the cell lines tested. Although the monotherapy of DNA-demethylating agents slightly suppressed the growth of normal hepatocytes, the addition of ALK TKIs did not increase the efficacy (Appendix A). In Karpas299 cells, the strongest synergistic effect was obtained with a combination of 10–20 nM alectinib and 1–2 μM AZA, 1.2–2.5 μM DAC, or 0.3–1.2 μM OR21. In SR-786 cells, the strongest synergistic effect was obtained with a combination of 40–160 nM alectinib and 0.5–2 μM AZA, 0.3–2.5 μM DAC, or 0.3–2.5 μM OR21. Since the pharmacological action of OR21 is derived from the release of DAC following OR21 processing, and OR21 showed less hematotoxicity than DAC in a mouse model, we chose to focus on AZA and OR21 in further experiments. We found that treating Karpas299 and SR-786 cells with a combination of alectinib and AZA or OR21 inhibited cell proliferation (Figure 3b) and induced apoptosis (Figure 3c) associated with the induction of cleaved caspase 3 (Appendix A) more strongly than treatment with each agent alone. However, no synergistic effect on the induction of cell cycle arrest was observed (Figure 3d). 

### 3.4. The DNA-Demethylating Agent and Alectinib Combination Induces Gene Expression Reprogramming in Karpas299 Cells

To elucidate the molecular mechanisms underlying the synergistic anti-tumor effect of the alectinib and DNA-demethylating agent combination therapy, we first confirmed the status of DNA methylation and STAT3 phosphorylation in ALCL cells. OR21 is a prodrug of DAC, with an efficacy that is almost identical to that of DAC in vitro. We found that treating Karpas299 cells with AZA or OR21 decreased the methylation status of *LINE1* repeats; however, the addition of alectinib did not increase DNA demethylation (Figure 4a). Similarly, although the treatment of Karpas299 cells with alectinib reduced the level of phosphorylated STAT3, the addition of alectinib with AZA or OR21 did not reduce STAT3 phosphorylation further (Figure 4b). 

To identify potential transcriptomic changes, we next analyzed RNA-seq data from Karpas299 cells treated with alectinib in combination with AZA or OR21. Although several ALK downstream target genes, such as IL-10, CD274, BCL2A1, etc., [35] were significantly down-regulated in Karpas299 cells after treatment with alectinib monotherapy, the addition of OR21 (Alec + OR21) did not enhance the effects (Appendix A). We found that 518 and 207 genes were up-regulated in Karpas299 cells following treatment with the combination of alectinib and OR21 or AZA, respectively (Figure 4c). Meanwhile, 675 and 533 genes were down-regulated by the combination of alectinib and OR21 or AZA, respectively (Appendix A). Thus, the combination of alectinib and OR21 induced gene expression reprogramming more strongly than the alectinib + AZA combination. Gene ontology analysis using DAVID showed that several gene sets, including those associated with the apoptotic signaling pathway, were up-regulated in alectinib + OR21-treated cells (Figure 4d).

Of the 518 genes up-regulated by the combination of alectinib and OR21, 243 genes (47%) were up-regulated specifically in response to the alectinib + OR21 combination, but not in response to alectinib or OR21 alone (Figure 4c). Thus, we speculated that these genes were implicated in key signaling pathways associated with the induction of the synergistic anti-ALCL effects of the alectinib + OR21 combination. The RNA-seq data analysis revealed that that the expression of *SFRP5*, *GPR171,* and *CEL* was dramatically induced in Karpas299 cells by the combination of alectinib and OR21 but not by each agent alone (Figure 4e). These findings were validated via real-time PCR and Western blot analysis (Supplementary Appendix A); a strong induction of SFRP5 by the combination treatment was observed only in Karpas299 cells among three cell lines. Although *CEL* gene expression was induced by the combination treatment in three cell lines, the protein expression was not detected in SR-786 cells. *GPR171* gene expression was induced by the combination treatment in three cell lines, but the expression was not detected via Western blot analysis. 

SFRP5 is a member of the secreted frizzled-related protein (SFRP) family and acts as a tumor suppressor and a secreted antagonist of the Wnt/β-catenin pathway [36]. The expression of SFRP proteins is silenced via promoter hypermethylation in several human malignancies [36,37], whereas treatment with DNA-demethylating agents restores their expression. We compared the DNA methylation status of *SFRP5* in normal CD3^+^ T cells and ALCL cells derived from patients with ALK^+^ or ALK^−^ ALCL using the GSE66881 dataset [19]. As expected, hypermethylation of the *SFRP5* promoter was observed in ALCL cells, regardless of ALK status (Appendix A). The activation of the Wnt/β-catenin pathway has been observed in ALK^+^ ALCL patients and ALK^+^ ALCL cell lines [38,39]. Furthermore, the knockdown of β-catenin using siRNA suppresses the growth of ALK^+^ ALCL cell lines [38]. Therefore, we hypothesized that combination treatment suppressed the Wnt/β-catenin pathway through the induction of SFRP5. As expected, treatment of Karpas299 cells with OR21 and alectinib reduced the amount of β-catenin associated with SFRP5 induction (Appendix A).

### 3.5. Combination Treatment with OR21 and Alectinib Suppresses Tumor Cell Growth in a Xenograft Mouse Model

Finally, we confirmed the efficacy of the combination of OR21 and alectinib in a Karpas299 xenograft mouse model (Figure 5a). Although each treatment alone had a small effect, the combination treatment significantly suppressed tumor cell growth (Figure 5b). All mice were sacrificed 13 days after the first drug administration, and xenograft tumors were isolated (Figure 5c). Again, the combination treatment showed a significant decrease in tumor weight (Figure 5d). On the other hand, combination treatment did not increase the drug-induced hematotoxicity compared with the effect of either single treatment (Figure 5e,f).

## 4. Discussion

The evolution of molecular targeted cancer therapy has improved treatment performance in several human malignancies. The ALK fusion gene (*NPM1-ALK*) was discovered in patients with ALCL in 1994 [3]. Since then, ALK TKIs, such as crizotinib and alectinib, which were developed for the treatment of ALK^+^ NSCLC patients, have been approved for ALK^+^ ALCL treatment in the US and Japan, respectively. However, the clinical benefits of ALK TKIs remain limited in ALK^+^ ALCL patients due to acquired drug resistance. 

In the present study, we demonstrated that DNA-demethylating agents, such as AZA, DAC, and OR21, enhanced the efficacy of alectinib when used in combination, compared to when each type of therapeutic agent was used alone. We have previously reported the development of the orally bioavailable DAC prodrug, OR21 [25,26]. Recently, the FDA approved ASTX727 [40], an oral formulation of a fixed-dose combination of DAC and cedazuridin (a cytidine deaminase inhibitor), for the treatment of adult patients with myelodysplastic syndromes and chronic myelomonocytic leukemia. OR21 is a silylated derivative of DAC, which is resistant to degradation by cytidine deaminase [25]. We previously showed that the efficacies of DAC and OR21 were very similar in adult T cell lymphoma, a mature T cell malignancy caused by infection with human lymphotropic virus I [26]. In accordance, the efficacies of DAC and OR21 were almost identical in the ALK^+^ ALCL cell lines tested in this study (Figure 1a,b). Another DNA-demethylating agent, AZA, suppressed cell growth in a dose-dependent manner, but had less effect on DNA demethylation than DAC or OR21 (Figure 1a,b). After being taken up by cells, only 10–20% of AZA is incorporated into DNA, while the majority is incorporated into RNA [41]. It is, therefore, possible that, in our study, AZA induced tumor growth inhibition via its incorporation into RNA. 

The synergistic efficacy of combining crizotinib with several drugs, including trametinib (an MEK inhibitor) [42], chloroquine [43], and DAC [42], has been reported. Since alectinib shows a better PFS and adverse event profile than crizotinib in Japanese patients with ALK-inhibitor-naïve ALK^+^ NSCLC [11], it has been approved for the treatment of recurrent or refractory ALK^+^ ALCL in Japan. In the present study, we therefore wished to determine whether the efficacy of alectinib could be further improved when used in combination with a DNA-demethylating agent. Although there were differences in the degree of effectiveness and optimal drug concentration, synergistic anti-ALCL effects were observed when alectinib was used with each of the DNA-demethylating agents tested (Figure 3a,b and Appendix A). Treatment with OR21 significantly changed gene expression, compared with AZA, both when used as a monotherapy and in combination with alectinib (Figure 4c and Appendix A). This may be because OR21 exhibits higher DNA-demethylating activity than AZA (Figure 1b). Based on these results, we think that the anti-ALCL efficacy of AZA could be derived from some non-epigenetic mechanism, distinct from that of DAC/OR21. Jin et al. showed that the combination of AZA and venetoclax synergistically promoted apoptosis via the induction of *NOXA* expression in a non-epigenetic manner [44]. We found that the combination of AZA and alectinib induced a synergistic effect on apoptosis in SR-786 cells but not in Karpas299 cells (Figure 3c).

To understand the molecular mechanisms underlying the induction of synergistic anti-ALCL effects, we focused on genes whose expression was up-regulated specifically by the combination of OR21 and alectinib, and identified *SRPF5*, *GPR171,* and *CEL* (Figure 4e). SFRP5 is a member of the SFRP family that acts as an antagonist of the Wnt/β-catenin pathway [36]. We confirmed that the combination treatment reduced the amount of β-catenin associated with SFRP5 induction in Karpas299 cells (Appendix A), suggesting that targeting the Wnt/β-catenin pathway through SFRP5 induction is one of the molecular mechanisms of synergistic anti-ALCL effects. However, since a strong induction of SFRP5 was only observed in Karpas299 cells (Appendix A), we think that other unknown mechanisms should also be involved (especially in cell lines other than Karpas299 cells). GPR171 is a G-protein-coupled receptor, which regulates feeding and anxiety behaviors by binding to its neuropeptide ligand, BigLen [45,46]. Of note, the interaction between BigLen and GPR171 suppresses T-cell-receptor-mediated signaling pathways and T cell expansion [47]. Although this is yet to be validated, the restoration of GPR171 expression might also contribute to the anti-ALCL effect of the combination of alectinib and OR21.

In a xenograft mouse model, the combination of OR21 and alectinib was shown to improve the efficacy. Importantly, the combination did not increase drug-induced hematotoxicity (Figure 5b–f). We also found that the combination did not increase the efficacy against normal hepatocytes (Appendix A). Based on these results, we think that the combination therapy enhances the anti-tumor effect without increasing the side effects on non-cancerous cells and that it is a feasible therapeutic approach for ALK^+^ ALCL.

## 5. Conclusions

In summary, we showed that the combination of alectinib and DNA-demethylating agents induced synergistic efficacy against ALK^+^ ALCL. Of these DNA-demethylating agents, OR21 exhibits a lower hematologic toxicity and can be safely administered over a long time period [26,28]. Moreover, a phase I clinical trial of OR21 in patients with high-risk myelodysplastic syndromes was recently initiated. Thus, alectinib and OR21 combination therapy may hold promise in improving treatment outcomes in patients with ALK^+^ ALCL.

## Figures and Tables

**Figure 1 cancers-15-05089-f001:**
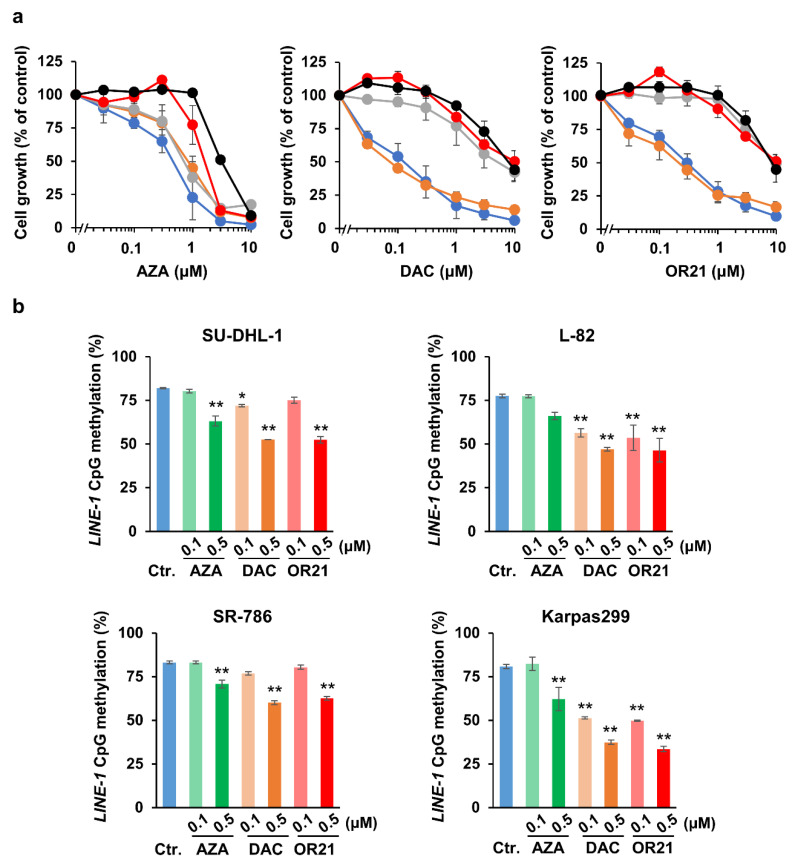
DNA-demethylating agents inhibit the proliferation of ALCL cell lines in vitro. (**a**) ALCL cell lines (SU-DHL-1: blue, L-82: orange, SR-786: gray, and Karpas299: red) and human primary hepatocytes (black) were treated with the indicated DNA-demethylating agents for 4 days. Cell proliferation was determined using the CCK-8 assay; the proliferation value of untreated cells was set at 100%. The results are expressed as the mean ± SD from three independent experiments. (**b**) The degree of *LINE-1* promoter methylation in each ALCL cell line after 3 days of treatment with each of the DNA-demethylating agents was measured using bisulfite pyrosequencing. The results are expressed as the mean ± SD of three independent experiments. The differences between the control and each treatment group were tested using the Dunnett test. * *p* < 0.05, ** *p* < 0.01.

**Figure 2 cancers-15-05089-f002:**
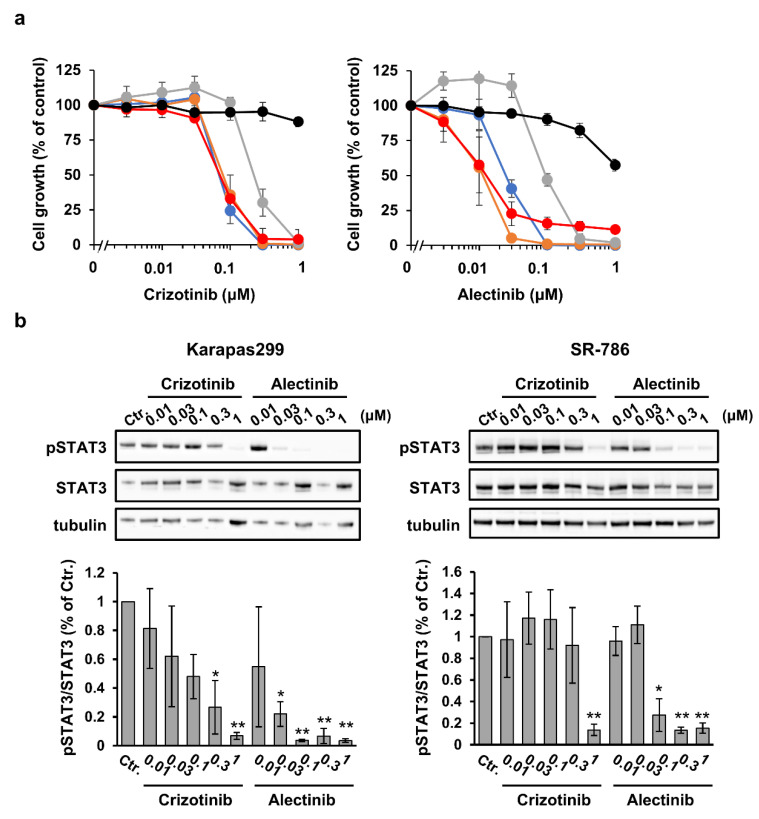
ALK TKIs inhibit the proliferation of ALCL cell lines in vitro. ALCL cell lines were treated with crizotinib or alectinib for 4 days. (**a**) The proliferation of the ALCL cells lines (SU-DHL-1: blue, L-82: orange, SR-786: gray, and Karpas299: red) and human primary hepatocytes (black) was determined using the CCK-8 assay; the proliferation value of untreated cells was set at 100%. The results are expressed as the mean ± SD of three independent experiments. (**b**) Immunoblots showing the phosphorylation status of STAT3 in cells exposed to the indicated treatment conditions. The signal intensities of phosphorylated STAT3 (Tyr705, pSTAT3) in the Western blotting results were measured and then divided by the intensity of STAT3. The uncropped blots are shown in File S1. The proportion of pSTAT3 in non-treated cells was defined as 100%. The results are the means of three independent experiments. The differences between the control and each treatment group were tested using the Dunnett test. * *p* < 0.05, ** *p* < 0.01.

**Figure 3 cancers-15-05089-f003:**
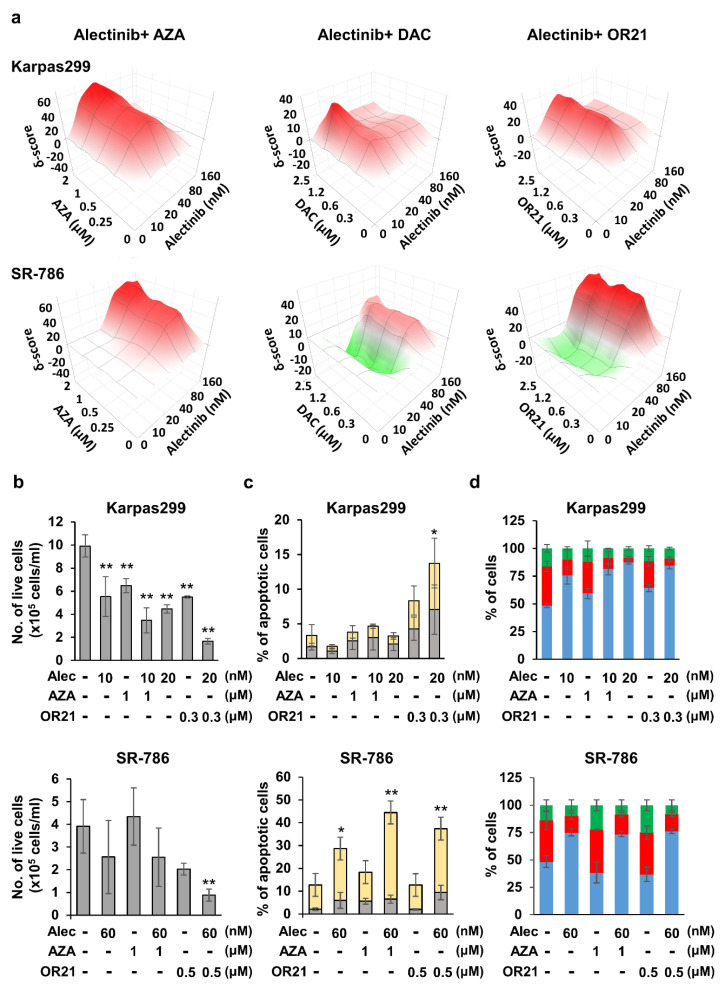
The synergistic inhibition of ALCL cell proliferation induced by the combination of alectinib and DNA-demethylating agents. Karpas299 and SR-786 cells were treated with each compound alone or in combination for 4 days. (**a**) The cell proliferation was determined using the CCK-8 assay. Synergy scores were calculated and synergy maps were generated using the SynergyFinder web application. ZIP Synergy scores of −10–10, >10, and <−10 indicate additive, synergistic, and antagonistic effects, respectively. (**b**) The number of live cells was measured via trypan blue staining. The results are expressed as the mean ± SD of three independent experiments. Differences between the control and each treatment group were tested using the Dunnett test. ** *p* < 0.01. (**c**) The number of cells stained with APC annexin V was measured via flow cytometry. Yellow and gray bars indicate annexin-V-positive/PI-negative cells (early apoptotic cells) and annexin-V-positive/PI-positive cells (late apoptotic cells), respectively. The results are expressed as the mean ± SD of three independent experiments. Differences between the control and each treatment group were tested using the Dunnett test. * *p* < 0.05, ** *p* < 0.01. (**d**) The cell cycle distribution was examined: blue columns, G1; red columns, S; green columns, G2/M. The results are expressed as the mean ± SD of three independent experiments.

**Figure 4 cancers-15-05089-f004:**
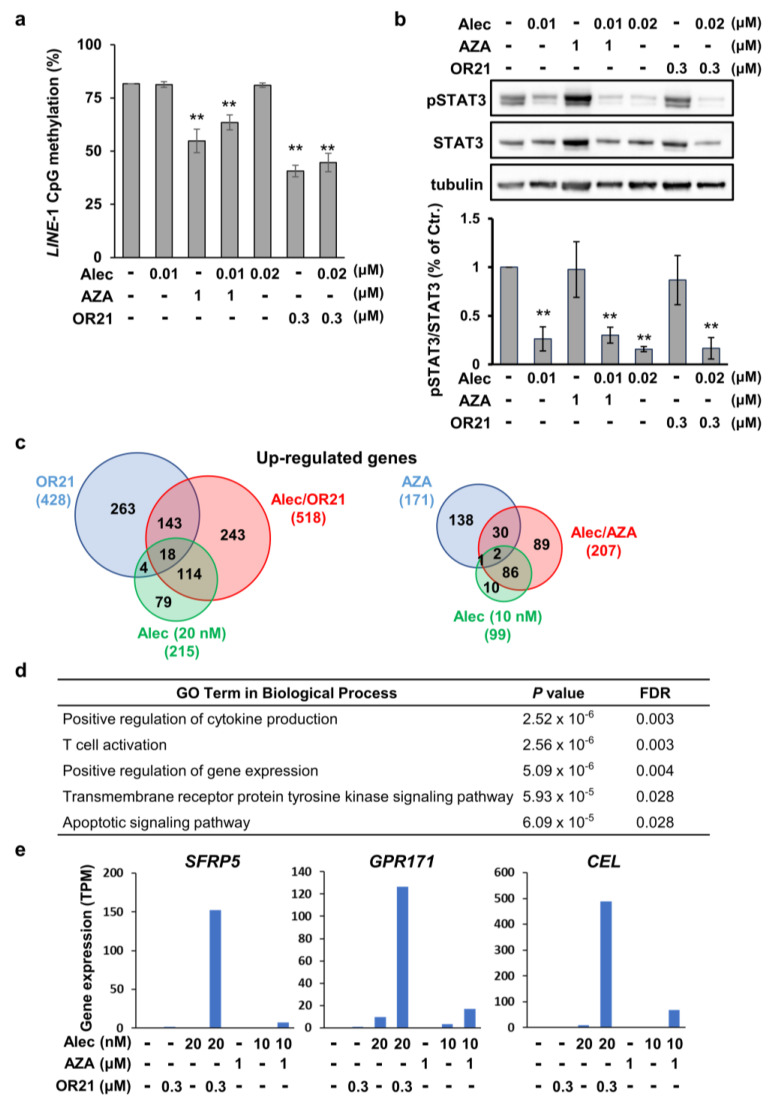
Global gene expression changes induced by the combination of alectinib and DNA-demethylating agents. Karpas299 cells were treated with each compound alone or in combination for 3 days. (**a**) The degree of *LINE-1* promoter methylation was determined via bisulfite pyrosequencing. The results are expressed as the mean ± SD of three independent experiments. ** *p* < 0.01. (**b**) Immunoblots showing the phosphorylation status of STAT3 in cells under different treatment conditions. The signal intensities of pSTAT3 in the Western blotting results were measured and then divided by the intensity of STAT3. The proportion of pSTAT3 in non-treated cells was defined as 100%. The results are the means of three independent experiments. Differences between the control and each treatment group were tested using the Dunnett test. The uncropped blots are shown in File S1. ** *p* < 0.01. (**c**) Venn diagrams showing the number of up-regulated (≥5 fold change) genes relative to the control. (**d**) The top 5 most significantly enriched GO terms obtained from the analysis of up-regulated genes in Karpas299 cells treated with the combination of alectinib with OR21. (**e**) Normalized expression levels (transcripts per million, TPM) of the *SFRP5*, *GPR171,* and *CEL* genes, obtained from the analysis of RNA-seq data.

**Figure 5 cancers-15-05089-f005:**
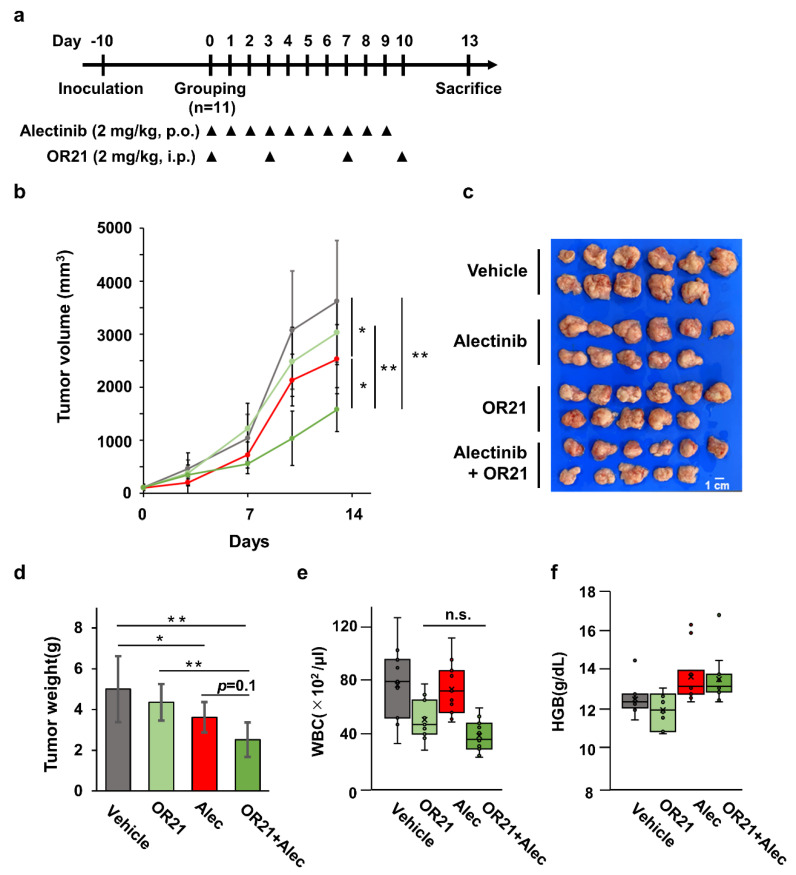
The anti-ALCL effect of OR21 and alectinib in a Karpas299 xenograft mouse model. (**a**) The experimental protocol for the establishment of the Karpas299 xenograft mouse model and treatment with alectinib and/or OR21. Karpas299 cells were subcutaneously inoculated into NOG mice and treated with vehicle (10% HP-β-CD containing 1% dimethyl sulfoxide, intraperitoneally, twice per week and 0.02N HCL, 10% DMSO, 10% CremophorEL, 15% PEG400, and 15% HP-β-CD, oral gavage, daily, *n* = 11), OR21 (2 mg/kg, intraperitoneally, twice a week, *n* = 11), alectinib (2 mg/kg, oral gavage, daily, *n* = 11), or OR21 + alectinib (2 mg/kg OR21 and 2 mg/kg alectinib, *n* = 11), as indicated by the arrowheads. (**b**) The mean of tumor volume in NOG mice was measured twice per week, starting on Day 0. Vehicle; grey, OR21; light green, alectinib; red, OR21+alectinib; dark green. (**c**) Tumors were isolated from mice sacrificed on Day 13. (**d**) The weight of the isolated tumors was measured. (**e**) The number of white blood cells (WBCs) and (**f**) the concentration of hemoglobin (HGB) from NOG mice 13 days after the first administration of compounds were determined. Differences between each group were tested using the Tukey test. * *p* < 0.05, ** *p* < 0.01, n.s.: no significance.

## Data Availability

RNA-seq data are available from DDBJ under accession number DRA017175. Other datasets generated and/or analyzed as part of this study are available from the corresponding author upon reasonable request.

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
