# Peer review of "A Combination of Alectinib and DNA-Demethylating Agents Synergistically Inhibits Anaplastic-Lymphoma-Kinase-Positive Anaplastic Large-Cell Lymphoma Cell Proliferation"

_cancers, 2023, doi:10.3390/cancers15205089_

Round 1

Reviewer 1 Report

With pleasure, I read the paper titled: “A combination of alectinib and DNA demethylating agents synergistically inhibits ALK-positive anaplastic large cell lymphoma cell proliferation” by Kawasoe and colleagues. Overall, the subject matter is of clinical interest to a wide array of readers. The topic is intellectually relevant to the journal Cancers. Collectively, the manuscript reads well, and data are summarized in pertinent figures. The main strength of the paper includes being among the first to examine the combination of TKI and DNA demethylating agent in ALK-positive ALCL. Nonetheless, the results are very brief, but sufficient as a preliminary report. However, additional work may be needed to make a solid story. I have the following comments/suggestions below.

Figures 1 and 2. Using the same concentrations of TKIs and DNA demethylating agents, have you examined the in-vitro safety of these drugs on non-cancerous (normal) cells, such as normal human kidney cells (HEK239) or normal fibroblast cells (HS68 and BJ)? It is recommended to quantify the pSTAT3 (normalized to STAT3) bands so you can provide proper results about the upregulation or downregulation of pSTAT3 protein.

Figure 3. Have you examined the synergy using additional cell lines besides Karpas299 cells to further validate your results? And whether additional cell lines were also used to examine the apoptosis and cell cycle assays.

Figure 4. It is recommended to quantify the pSTAT3 (normalized to STAT3) bands so you can provide proper results about the upregulation or downregulation of pSTAT3 protein.

General questions that will substantially enhance the quality of your research … It is would be great to validate the annexin V flowcytometry results using western blot and probing for some apoptotic markers, such as cleaved-caspase 3 and cleaved-PARP? Have you tried to examine if ALK-negative ALCL cells are less sensitive to TKI and DNA demethylating agents compared with ALK-positive ALCL cells? Does genetic inhibition of ALK (using knockdown or knockout) synergize pharmacologically with DNA demethylating agent? The study will become more significant if in-vivo xenograft data are included; particularly if you could examine the potential synergy between TKI and DNA demethylating agents. Regarding the RNA-seq data, did it show enhanced downregulation in ALK downstream target genes for the combination agents versus single agents? The combination therapy should be tested on normal cells as well to explore safety on non-cancer cell lines. Does overexpression of ALK in ALCL cells enhance the sensitivity to combination of TKI and DNA demethylating agents? More work is needed to delineate the mechanism of reduced proliferation of ALCL cells upon combination of TKI and DNA demethylating agent.

It is fine. Some minor English polishing.

Author Response

We greatly appreciate your valuable comments.

They helped us to improve our manuscript.

According to your suggestions, we conducted additional experiments.

Please find our responses to each comment below.

[Reviewer 1 (R1)-Comment 1 (C1)] Figures 1 and 2. Using the same concentrations of TKIs and DNA demethylating agents, have you examined the in-vitro safety of these drugs on non-cancerous (normal) cells, such as normal human kidney cells (HEK239) or normal fibroblast cells (HS68 and BJ)?

[Response to R1-C1] According to the comments, we performed a cell growth assay using human primary hepatocytes, which we have used previously (Cancers 2021, 13(21), 5413). We replaced Figure 1a and 2a with new experimental data. As shown in Figure 1a, treatment of hepatocytes with AZA, DAC, and OR21suppressed cell growth to some extent, possibly because the cells proliferated slowly. On the other hands, ALK TKIs did not inhibit their growth as shown in Figure 2a.

To explain these data, we added sentences in Results as follows;

[Page 8, Lines 218-220] Treatment of primary human hepatocytes with AZA, DAC, and OR21suppressed cell growth to some extent, possibly because the cells proliferated slowly (Fig. 1a).

[Page 9, Line 234] Both of crizotinib and alectinib had little efficacy against normal hepatocytes (Fig. 2a).

[R1-C2] It is recommended to quantify the pSTAT3 (normalized to STAT3) bands so you can provide proper results about the upregulation or downregulation of pSTAT3 protein.

[Response to R1-C2] According to the comments, we quantified the signal intensities of pSTAT3 and STAT3 and replaced Figure 2b and Figure 4b with the quantification data.

[R1-C3] Figure 3. Have you examined the synergy using additional cell lines besides Karpas299 cells to further validate your results? And whether additional cell lines were also used to examine the apoptosis and cell cycle assays.

[Response to R1-C3] We tested four cell lines (Karpas299, SR-786, SU-DHL1 and L-82) to determine the synergy score as shown in Figure 3a and Supplementary Figure 1a. Following the reviewer’s comments, we performed additional validation experiments on the apoptosis and cell cycle assay using SR-786 cells. We replaced Figure 3 with new experimental data. Please refer to Figure 3a-d and Supplementary Figure 1a for details.

To explain these data, we added and modified sentences in Results as follows;

[Page 9, Line 252- Page 10, Line 254] In SR-786 cells, the strongest synergistic effect was obtained with a combination of 40–160 nM alectinib and 0.5–2 μM AZA, 0.3–2.5 μM DAC, or 0.3–2.5 μM OR21.

[Page 10, Lines 256-259] We found that treating Karpas299 and SR-786 cells with a combination of alectinib and AZA or OR21 inhibited cell proliferation (Fig. 3b) and induced apoptosis (Fig. 3c) associated with induction of cleaved caspase 3 (Supplementary Fig. 2) more strongly than with each agent alone.

[R1-C4] Figure 4. It is recommended to quantify the pSTAT3 (normalized to STAT3) bands so you can provide proper results about the upregulation or downregulation of pSTAT3 protein.

[Response to R1-C4] As we mentioned in [Response to R1-C2], we quantified the signal intensities of pSTAT3 and STAT3 and replaced Figure 4b with the quantification data.

[R1-C5] It would be great to validate the annexin V flowcytometry results using western blot and probing for some apoptotic markers, such as cleaved-caspase 3 and cleaved-PARP?

[Response to R1-C5] Following the comments, we performed additional western blot analysis using anti-cleaved caspase 3 antibody and generated an additional figure (Supplementary Fig. 2). The combination treatment increased the amount of cleaved caspase 3 to a greater extent compared to the effects of either single treatment. Please refer to Supplementary Figure 2 for details.

To explain these data, we added sentences in Results as follows;

[Page 10, Lines 255-258] We found that treating Karpas299 and SR-786 cells with a combination of alectinib and AZA or OR21 inhibited cell proliferation (Fig. 3b) and induced apoptosis (Fig. 3c) associated with induction of cleaved caspase 3 (Supplementary Fig. 2) more strongly than with each agent alone.

[R1-C6] Have you tried to examine if ALK-negative ALCL cells are less sensitive to TKI and DNA demethylating agents compared with ALK-positive ALCL cells?

[Response to R1-C6] We could not test for ALK-negative ALCL since these are not commercially available. However, we found that cell lines established from breast implant-associated ALK-negative anaplastic large cell lymphoma (TLBR-1, 2 and 3) in DSMZ. Since the cell lines were not conventional ALK-negative ALCL cases, we do not include the data in this manuscript. However, we would like to test the efficacy of DNA demethylating agents against TLBR cell lines in the next future project. Thank you very much for your valuable comments.

[R1-C7] Does genetic inhibition of ALK (using knockdown or knockout) synergize pharmacologically with DNA demethylating agent?

[Response to R1-C7] Due to low knockdown efficiency (infection efficiency with lentivirus containing sh-RNA targeting ALK gene), we could not add the validation experiments. But we think if we could effectively knockdown ALK fusion gene, the treatment with DNA demethylating agents would improve the anti-ALCL effects.

[R1-C8] The study will become more significant if in-vivo xenograft data are included; particularly if you could examine the potential synergy between TKI and DNA demethylating agents.

[Response to R1-C8] In accordance with the comments, we performed additional in vivo xenograft mouse experiments. The combination of alectinib with OR21 suppressed tumor cell growth to a greater extent than the effect of either single treatment. Please refer to Figure 5 for details.

To explain these data, we added a paragraph in Results (Page 11, Line 309- Page 12 line 317).

[R1-C9] Regarding the RNA-seq data, did it show enhanced downregulation in ALK downstream target genes for the combination agents versus single agents?

[Response to R1-C9] Following the comments, we re-examined the RNA-seq data and added the data in Supplementary Figure 3a. Although several ALK downstream target genes such as IL-10, CD274, BCL2A1, etc. were significantly down-regulated by alectinib monotherapy, the addition of OR21 (Alec+OR21) did not increase the effects. Please refer to Supplementary Figure 3a for details.

To explain these data, we added sentences in Results as follows;

[Page 10, Lines 273-276] Although several ALK downstream target genes such as IL-10, CD274, BCL2A1, etc. were significantly down-regulated in Karpas299 cells after treatment with alectinib monotherapy, the addition of OR21 (Alec+OR21) did not enhance the effects (Supplementary Fig. 3a).

[R1-C10] The combination therapy should be tested on normal cells as well to explore safety on non-cancer cell lines.

[Response to R1-C10] We agree with the reviewer’s concern. Following the comments, we have performed a cell growth assay using normal human hepatocytes treated with DNA demethylating agents and ALK TKIs and added these data in Supplementary Figure 1b. As we showed in revised Figure 1a, monotherapy of DNA demethylating agents suppressed the growth (please refer to [Response to R1-C1]). However, the addition of ALK TKIs did not increase efficacy (Supplementary Figure 1b). We also conducted in additional in vivo xenograft mouse experiments (please refer to [Response to R1-C8]) and found that the combination of OR21 with allectinib did not increase hematotoxicity compared to either single treatment alone (Fig.5e,f). Based on these results, we think that the combination therapy enhances anti-tumor effects without increasing side effects on non-cancerous cells.

To describe these points, we added the following sentence in Results and Discussion;

[Page 9, Lines 249-250] Although the monotherapy of DNA demethylating agents slightly suppressed the growth of normal hepatocytes, the addition of ALK TKIs did not increase the efficacy (Supplementary Fig. 1b).

[Page 13, Lines 371-376] In a xenograft mouse model, the combination of OR21 and alectinib showed improved the efficacy. Importantly, the combination did not increase drug-induced hematotoxicity (Fig. 5b-f). We also found that ALK TKIs did not increase the efficacy against DNA demethylarting agents treated-normal hepatocyte (Supplementary Fig. 1b). Based on these results, we think that the combination therapy enhances the anti-tumor effect without increasing the side effects on non-cancerous cells and it is a feasible therapeutic approach for ALK+ALCL.

[R1-C11] Does overexpression of ALK in ALCL cells enhance the sensitivity to combination of TKI and DNA demethylating agents?

[Response to R1-C11] I understand the reviewer’s concern, but as we mentioned in [Response to R1-C6], we do not have ALK-negative ALCL cell lines. Therefore, we do not have an answer to this question at this time. We’d like to use breast implant-associated ALK-negative anaplastic large cell lymphoma cell lines (TLBR-1, 2 and 3) to confirm the question in the next future experiments.

[R1-C12] More work is needed to delineate the mechanism of reduced proliferation of ALCL cells upon combination of TKI and DNA demethylating agent.

[Response to R1-C12] According to the comments, we performed additional western blot analysis. Since we found that the combination treatment induced gene expression of SFRP5, which is known as a secreted antagonist of the Wnt/β-catenin pathway, we focused on SFRP5. We found that the combination induced SFRP5 and reduced of β-catenin in Karas299 cells as shown in Supplementary Figure 6. Since strong induction of SFRP5 was observed only in Karpas299 cells, we think that induction of SFRP5 is one of the anti-ALCL mechanisms and that other mechanisms should also be involved (especially in cell lines other than Karpas299 cells).

To describe these points, we added the following sentence in Results and Discussion;

[Page 11, Lines 301-307] The activation of the Wnt/β-catenin pathway has been observed in ALK+ ALCL patients and ALK+ ALCL cell lines[38, 39]. Furthermore, the knockdown of β-catenin using siRNA suppresses the growth of ALK+ ALCL cell lines[38]. Therefore, we hypothesized that combination treatment suppressed Wnt/β-catenin pathway through induction of SFRP5. As expected, treatment of Karpas299 cells with OR21 and alectinib reduced the amount of β-catenin associated with SFRP5 induction (Supplementary Fig. 6).

[Page 13, Lines 360-366] SFRP5 is a member of the SFRP family, which acts as an antagonist of the Wnt/β-catenin pathway[36]. We confirmed that the combination treatment reduced the amount of β-catenin associated with SFRP5 induction in Karas299 cells (Supplementary Fig. 6), suggesting that targeting Wnt/β-catenin pathway through SFRP5 induction is one of the molecular mechanisms of synergistic anti-ALCL effects. However, since strong induction of SFRP5 was only observed in Karpas299 cells (Supplementary Fig. 4a), we think that other unknown mechanisms should also be involved (especially in cell lines other than Karpas299 cells).

Reviewer 2 Report

In this manuscript, the authors investigate the effect of a combination of alectinib and DNA demethylating agents to cell proliferation. The authors present some results to suggest that the drug combination synergistically suppressed cell growth via reprogramming the gene expression.  I think some concerns should be addressed before publication in Cancers.

Specific comments:

1.     In Figure 1b, after different agents treatment, the methylation level was reduced, I suggest the authors conduct a P-value assay of methylation level among control and different agents treatment.

2.     In figure 2b, to better confirm the results, whether the authors can quantify the western blot bands of pSTAT3 and conduct a P-Value assay?

3.     In figure 4a, add P-Value assay, figure 4b, please quantify the he western blot bands of pSTAT3 and conduct a P-Value assay. Figure 4e, I suggest the authors add the protein expression level of the three genes.

Author Response

We greatly appreciate your valuable comments.

They helped us to improve our manuscript.

According to your suggestions, we conducted additional experiments.

Please find our responses to each comment below.

[R2-C1] In Figure 1b, after different agents treatment, the methylation level was reduced, I suggest the authors conduct a P-value assay of methylation level among control and different agents treatment.

[Response to R2-C1] In response to the comments, we conducted a statistical analysis and replaced Figure 1b.

[R2-C2] In figure 2b, to better confirm the results, whether the authors can quantify the western blot bands of pSTAT3 and conduct a P-Value assay?

[Response to R2-C2] As we mentioned in [Response to R1(Reviewer 1)-C2], we quantified the signal intensities. Following the comments, we also conducted a statistical analysis.

[R2-C3] In figure 4a, add P-Value assay, figure 4b, please quantify the western blot bands of pSTAT3 and conduct a P-Value assay. Figure 4e, I suggest the authors add the protein expression level of the three genes.

[Response to R2-C3] According to the comments, we quantified signal intensities in Western blot experiments and performed statistical analysis. Please refer to the replaced Figure 4a, b. We also performed additional Western blot analysis using anti-SFRP5, anti-GPR171, and anti-CEL antibodies to address the comments. Please refer to Supplementary Figure 4b for details.

Reviewer 3 Report

This manuscript explores the potential of combining alectinib, a drug commonly used to treat ALK-positive anaplastic large cell lymphoma (ALCL), with DNA demethylating agents, specifically OR21, to improve treatment outcomes. The authors conducted in vitro experiments to demonstrate the synergistic effects of this combination therapy on ALK+ ALCL cells, and investigated the underlying mechanisms of this synergy, including changes in gene expression. The manuscript concludes that the combination of alectinib and OR21 has promising potential in improving treatment outcomes for ALK+ ALCL patients. However, there are some questions still needed to be answered:

1. In vivo studies: While the authors demonstrated the synergistic effect of alectinib and OR21 in vitro, additional studies in animal models could help establish the efficacy of this combination therapy in vivo.

2. Mechanistic studies: While the authors demonstrated that the combination of alectinib and OR21 induced apoptosis and cell cycle arrest in ALK+ ALCL cells, the underlying molecular mechanisms for this effect are not fully understood. While the authors identified three genes through RNA-seq analysis that may play a role in the synergistic anti-ALCL effects of the combination therapy, additional mechanistic studies could provide a deeper understanding of the signaling pathways and molecular targets involved.

3. Some figures are missing p-values, which should be included for clarity and rigor.

Author Response

We greatly appreciate your valuable comments.

They helped us to improve our manuscript.

According to your suggestions, we conducted additional experiments.

Please find our responses to each comment below.

[R3-C1] In vivo studies: While the authors demonstrated the synergistic effect of alectinib and OR21 in vitro, additional studies in animal models could help establish the efficacy of this combination therapy in vivo.

[Response to R3-C1] As we mentioned in [Response to R1(Reviewer 1)-C8], we performed additional in vivo experiments. Please refer to [Response to R1-C8] and Figure 5 for details.

[R3-C2] Mechanistic studies: While the authors demonstrated that the combination of alectinib and OR21 induced apoptosis and cell cycle arrest in ALK+ ALCL cells, the underlying molecular mechanisms for this effect are not fully understood. While the authors identified three genes through RNA-seq analysis that may play a role in the synergistic anti-ALCL effects of the combination therapy, additional mechanistic studies could provide a deeper understanding of the signaling pathways and molecular targets involved.

[Response to R3-C2] We agree with the reviewer’s concern. As we mentioned in [Response to R1-C12], we focused on the induction of SFRP5 to understand the mechanisms of combination effects. We performed additional experiments and found the combination treatment reduced the amount β-catenin. Thus, up-regulation of SFRP5 is one of the molecular mechanisms to induce the combination effects. Please refer to [Response to R1-C12] and Supplementary Figure 6 for details.

[R3-C3] Some figures are missing p-values, which should be included for clarity and rigor.

[Response to R3-C3] According to the comments, we have performed statistical analysis and added P-values in replaced Figures (Figure 1a, 2b, 4a, and 4b).

Round 2

Reviewer 1 Report

For the most part, the authors addressed the raised concerns experimentally and textually, if experiments were not done. Additionally, the authors acknowledged the limitations and highlighted them as a future investigation. No manuscript is complete. Nonetheless, the present manuscript has sufficient data to support the generated claims, and the content is scientifically valid. All in all, I congratulate the authors for a well-done job and I endorse acceptance of manuscript in its current form.

Minor English editing is needed

Reviewer 2 Report

Accept